# SpEmoC: Large-Scale Multimodal Dataset for Speaking Segment Emotion Insights

## Abstract

Understanding human emotions in spoken conversations is a key challenge in affective computing, with applications in empathetic AI, human-computer interaction, and mental health monitoring. Existing datasets lack scale, tightly aligned modalities, and balance in emotion diversity, thereby limiting robust multimodal models. To address this, we propose **SpEmoC**, a large-scale **Sp**eaking segment **Emo**tion dataset for **C**onversations. SpEmoC comprises 306,544 clips from 3,100 English-language videos, featuring synchronized visual, audio, and textual modalities annotated for seven emotions, and yields a refined set of 30,000 high-quality clips. It focuses on speaking segments under diverse conditions like low lighting and resolution, with a threshold-based filtering and human annotation ensuring a balanced dataset. SpEmoC is class-balanced, which enables fair learning across all emotions and leads to comparably balanced performance across all classes. We introduce a lightweight CLIP-based baseline model with a fusion network and a novel multimodal contrastive loss to enhance emotion alignment. We conduct a series of experiments demonstrating strong results, establishing SpEmoC as a reliable benchmark for advancing multimodal emotion recognition research.

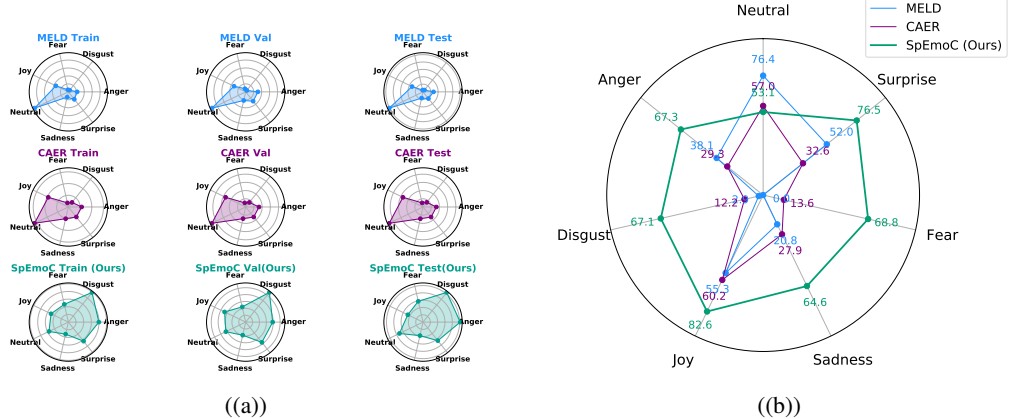

((a))                              ((b))

Figure 1: (a) Emotion class distribution comparison between **SpEmoC**, MELD, and CAER datasets across train, test, and validation splits. The proposed SpEmoC dataset shows a more balanced distribution across all seven emotion classes, whereas the MELD Poria et al. (2018) and CAER Lee et al. (2019) datasets are skewed toward the Neutral class. Subfigure (b) Class-wise recognition performance of the baseline model on SpEmoC. Unlike prior datasets where minority emotions (e.g., fear, disgust) are poorly recognized, the balanced distribution of SpEmoC enables comparably robust F1-score across all emotion classes.

## 1 Introduction

Understanding human emotion from multimodal cues is a fundamental task in affective computing, with wide-ranging applications in human-computer interaction, mental health, pain detection for medical diagnostics Lucey et al. (2011), and social robotics Picard (2000). Emotion is inherently multimodal, manifested through facial expressions, speech prosody, and linguistic content, making the integration of visual, audio, and textual modalities essential for comprehensive emotion

recognition. Recent advances in large-scale pretrained models, such as CLIP Radford et al. (2021), wav2vec Baevski et al. (2020), and BERT Devlin et al. (2018), have significantly enhanced unimodal feature extraction, yet leveraging these models for fine-grained, multimodal emotion understanding in real-world conversational settings remains underexplored.

Multimodal Emotion Recognition (MER) faces significant challenges that hinder its deployment in dynamic, dialogue-driven contexts. Most existing benchmarks focus on unimodal settings, such as facial expressions (Li et al. (2017); Zeng et al. (2018)) or audio (Schuller et al. (2011); El Ayadi et al. (2011)). They suffer from limited modality alignment and annotation scale (Poria et al. (2017a); Zadeh et al. (2018); Albanie et al. (2018)). Datasets like the Multimodal EmotionLines Dataset (MELD) Poria et al. (2018), with 13,000 utterances from the TV series *Friends*, and CAER Lee et al. (2019), which includes 13,201 video clips from 79 TV shows with audio and visual tracks, offer multimodal annotations at limited scales. Similarly, EmoWOZ Feng et al. (2021) provides 11,000+ task-oriented dialogue utterances with multimodal labels. However, these datasets are constrained by relatively small sizes and imbalanced emotion distributions, with MELD and CAER dominated by "Neutral" emotions and underrepresentation of "Fear" and "Disgust" (see Figure 1). These datasets lack real-world diversity and, being built from TV series, often reuse characters across splits, making test sets not truly unseen. Similarly, the M3ED dataset Zhao et al. (2022) offers 9,000 utterances from TV series but falls short in capturing the breadth of emotional expressions needed for generalization. Furthermore, the reliance on expensive human annotations and the absence of synchronized multimodal alignment limit the scalability and applicability of these datasets. Fusing heterogeneous modalities also remains challenging due to differences in data representation, temporal dynamics, and emotional relevance across text, audio, and visual streams (Zadeh et al. (2018); Wollmer et al. (2013)).

To address these limitations, we introduce **SpEmoC** : a large-scale **Sp**eaking segment **Emo**tion dataset for **C**onversations designed to support emotion recognition in real-world, multimodal interactions. SpEmoC comprises 30,000 refined clips, curated from 306,544 raw video segments sourced from 3,100 English-language movies and TV series across diverse genres, including drama, comedy, horror, thriller, romance, and history. This dataset captures a wide range of emotional expressions in naturalistic settings, with temporally aligned video, audio, and text data. This enables the study of cross-modal emotion alignment and fusion. Notably, SpEmoC is balanced across all seven emotion classes through targeted filtering and refinement, as illustrated in Figure 1. Inspired by recent efforts like EmotionCLIP Zhang et al. (2023), which utilizes large-scale TV series data for emotion representation learning, SpEmoC significantly improves scale, diversity, and real-world applicability for conversational MER.Our contributions are four-fold:

- **SpEmoC Dataset.**We introduce SpEmoC, a large-scale multimodal dataset with 30,000 temporally aligned video, audio, and text clips from 3,100 movies and TV series. Unlike prior datasets, SpEmoC provides a balanced distribution across all seven emotion classes. This enables robust recognition not only of the dominant classes (e.g., *Neutral*) but also of underrepresented ones such as *Fear* and *Disgust*.

- **Automatic Annotation Pipeline.** We propose a scalable annotation methodology that employs pretrained emotion recognition models for text and audio to generate emotion labels, using a fusion algorithm based on emotion logits to infer video-level emotions.

- **Multimodal Contrastive Loss.** We develop an Extended Re-weighted Multimodal Contrastive Loss (ERMC), enhanced with KL-divergence-based weighting, to align emotional embeddings across modalities using predicted unimodal sentiment distributions.

- **Efficient Baseline Model.** We propose a lightweight model integrating pretrained CLIP encoders for video and text, a compact HuBERT-based audio encoder, and a fusion MLP classifier, achieving strong performance with minimal trainable parameters. Despite its compact size, the model achieves balanced per-class accuracy.

SpEmoC significantly improves the scale, diversity, and real-world applicability of conversational MER, and by mitigating class imbalance through targeted filtering, it enables models to learn previously underrepresented emotions such as fear and disgust (see Figure 1 (a)). As demonstrated by the balanced distribution in Figure 1 (b), SpEmoC provides a stronger benchmark for advancing multimodal emotion recognition, with potential for balanced performance across classes pending further evaluation. Together, **SpEmoC**, our annotation pipeline, and our baseline model provide

a foundation for scalable, weakly supervised, and modality-aware emotion recognition, paving the way for future research in affective computing.

## 2 RELATED WORK

Emotion recognition research relies on multimodal datasets, each enhancing understanding of emotional expressions across modalities. IEMOCAP Busso et al. (2008) offers 10,039 utterances with audio, video, and motion data for categorical and dimensional affect, while MELD Poria et al. (2018) provides 13,000 utterances from *Friends* with text, audio, and visual annotations using Ekman's classes. CAER Lee et al. (2019) includes 13,201 video clips from 79 TV shows, manually annotated for seven emotions with context emphasis, whereas RAVDESS Livingstone & Russo (2018) delivers 7,356 audio-video files of scripted emotions, and EmoReact Nojavanasghari et al. (2016) features 1,102 clips of children's reactions for six emotions. These datasets, though foundational, are limited by scale, diversity, and consistency, hindering broader applicability. Recent advances include CMU-MOSEI Zadeh et al. (2018) with 22,856 video segments for monologue sentiment, AffWild2 Kollias & Zafeiriou (2020) with 564 in-the-wild valence-arousal annotations, EmoWOZ Feng et al. (2021) with 11,000+ dialogue utterances, and M3ED Zhao et al. (2022) with 9,000 TV series utterances, evolving from controlled settings (IEMOCAP Busso et al. (2008)) to naturalistic ones with CMU-MOSEI Zadeh et al. (2018), PanoSent Luo et al. (2024), and MELD Poria et al. (2018) across text, speech, and visual cues.

Further progress is evident with newer datasets and model-level innovations. EmotionTalk Sun et al. (2025), a Chinese multimodal dataset with 19,250 utterances and rich annotations, and EMOVOME Gómez-Zaragozá et al. (2024), featuring 999 spontaneous Spanish voice messages, enhance cross-lingual and real-world diversity. EmotionLLAMA's MERR Cheng et al. (2024) offers 28,618 coarse-grained and 4,487 fine-grained samples, while EmotionCLIP Zhang et al. (2023) leverages large-scale TV series data for emotion representation learning, advancing annotation scalability. At the model level, the Multimodal Transformer (MuIT) Tsai et al. (2019) integrates cross-modal attention, the Dynamic Fusion Graph Network (Chen & Shi (2025); Wang et al. (2025); Zhao et al. (2025)) models contextual relationships, and Contrastive Emotion Alignment (Zhang et al. (2025); Wu et al. (2025)) aligns multimodal embeddings for robustness. Despite these efforts, challenges in data scale, annotation scalability, and handling real-world noise persist, motivating the development of SpEmoC as a larger, more diverse, and validated resource.

Table 1: Comparison with existing multimodal emotion recognition datasets. Modalities: A = Audio, V = Visual, T = Text.

| Dataset | Samples | Modalities | No. of Emotions | Source |
|---|---|---|---|---|
| IEMOCAP Busso et al. (2008) | 10,039 | V, A, T, Motion | 8 | Acted dialogues |
| MELD Poria et al. (2018) | 13,000 | V, A, T | 7 | Friends TV show |
| CAER Lee et al. (2019) | 13201 | V, A | 7 | TV shows |
| RAVDESS Livingstone & Russo (2018) | 7,356 | V, A | 8 | Studio-acted clips |
| EmoReact Nojavanasghari et al. (2016) | 1,102 | V, A | 8 | YouTube videos |
| **SpEmoC(Proposed)** | **306,544** (raw), **30,000** (refined) | **V, A, T** | **7** | **Movies & TV series** |

## 3 SpEmoC DATASET CONSTRUCTION

We introduce **SpEmoC**, a large-scale multimodal dataset for emotion recognition, focused on **speaking segments** and containing synchronized **video**, **text**, and **audio** samples. Clips are extracted from 3,100 publicly available English-language movies and TV series, capturing natural, emotionally diverse content across genres (drama, comedy, horror, etc.), formats (color, black-and-white), and conditions (low-light, varying resolutions). SpEmoC preserves authentic speech and facial expressions while avoiding dubbed content and subtitles while reducing cultural bias. We showcase the diversity of our dataset in terms of visual style and emotional expression in Figure 2. A detailed explanations provided in the **Appendix** A

**Motivation**

Existing datasets often lack scale, synchronization, and emotional diversity. Most of them provide either short, caption-based content or acted emotions recorded in constrained settings, and they also suffer from severe class imbalance, with the *neutral* class heavily overrepresented and minority

emotions (e.g., *fear*, *disgust*) largely neglected. In contrast, SpEmoC offers real-world emotion-rich scenarios with tightly aligned modalities and a balanced distribution across all emotion classes. It is designed to support the development of robust, generalizable emotion recognition models that can learn from vocal tone, facial expressions, and contextual language. Table 2 provides an overview of the proposed SpEmoC dataset for multimodal emotion recognition, detailing its source, structure, modalities, annotation process, splitting policy, and emotion coverage, while Table 1 compares it with existing multimodal emotion recognition datasets.

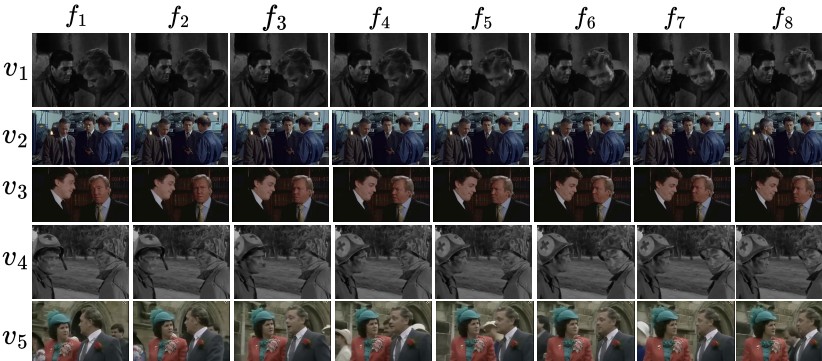

Figure 2: Examples from SpEmoC showing variation in genre, lighting, color, and expression. Each row displays 8 sampled frames from a distinct clip.

## 3.1 DATA COLLECTION AND PROCESSING PIPELINE

We present a scalable multi-stage pipeline that processes long videos into synchronized multimodal emotion clips. Let $V = \{V_1, V_2, \ldots, V_{3100}\}$ be the set of source videos, each $V_k$ with duration $t_k \geq 40$ minutes.

**1. Dialogue Segmentation:** We use the Whisper ASR model Radford et al. (2023) to transcribe each video with word-level timestamps. Segments are retained if they: (i) contain at least 12 words, ensuring sufficient context, and (ii) ended with terminal punctuation (e.g., ., !, ?) Each video $V_k$ yields segments $\{v_{k_1}, v_{k_2}, \ldots\}$, each defined by start/end times and transcript $T_{k_i}$, totaling:

$$N_{\text{clips}} = \sum_{k=1}^{3100} \sum_{i=1}^{m_k} 1 \approx 306{,}544. \tag{1}$$

**2. Multimodal Extraction:** For each segment $v_{k_i}$, we extract:

- **Text:** $T_{k_i}$, the transcribed dialogue text, directly obtained from the Whisper model.

- **Audio:** $A_{k_i}$, from the interval $[t_{\text{start}}^{k_i}, t_{\text{end}}^{k_i}]$, using FFmpeg Developers (2025). Duration of audio clip: $\Delta t^{k_i} = t_{\text{end}} - t_{\text{start}}$. Where $t_{\text{start}}$ is start time and $t_{\text{end}}$ is end time.

- **Visual:** $v_{k_i}$, video segment from the same interval $[t_{\text{start}}^{k_i}, t_{\text{end}}^{k_i}]$, with frame count: $N_{\text{frames}}^{k_i} \approx \Delta t^{k_i} \times$ FPS. This results in 30 million video frames across all clips.

- **Human and Face detection:** To focus analysis on subjects in the clips, each frame is processed with YOLOv8 Varghese & Sambath (2024) to detect *human* and *face* bounding boxes. We retain only these regions, ensuring consistent alignment across modalities and directing the model's attention to subjects' facial and bodily cues. The bounding box co-ordinates are defined as $B_{j,k} = [x_{\text{min}}, y_{\text{min}}, x_{\text{max}}, y_{\text{max}}]$, where $B_{j,k}$ denotes the box for the $j$-th person in the $k$-th frame. By emphasizing subject-specific regions, this strategy improves synchronization of audio, text, and visual cues, thereby enhancing recognition of person-centered emotions.

**3. Synchronization Check:** We verify alignment between modalities using timestamp consistency:

$$|\text{Duration}(A_k) - (t_{\text{end}}^k - t_{\text{start}}^k)| \leq \epsilon, \quad \text{within a tolerance} \quad \epsilon = 0.1 \text{ sec.} \tag{2}$$

Clips failing this check are reprocessed or flagged.

**4. Metadata Generation:** For each clip $c_k = \{v_k(t), A_k(t), T_k(t)\}$, we save metadata including timestamps, file paths, and transcription in a JSON file to support downstream annotation and training.

This pipeline yields a high-quality, synchronized multimodal dataset optimized for fine-grained emotion understanding across realistic scenarios.

Table 2: Overview of the Proposed Multimodal Emotion Recognition SpEmoC Dataset, Highlighting Source, Structure, Modalities, Annotation, Splitting policy, and Emotion Coverage.

| **Data Source and Composition** | |
|---|---|
| Source | YouTube (Movies and TV Series) |
| Number of Videos | 3,100 |
| Video Length | $\geq 40$ minutes each |
| Video Types | Color and Black-and-White , English language , Non-dubbing, Subtitle independent |
| Video genres | Drama, Comedy, Horror, Thriller, Romance, History, etc. |
| Total Number of Clips | 306,544 |
| Average Clip Duration | 3–6 seconds |
| Total Frames | 30 million+ |
| Focus Per Clip | Speaking segments only |
| **Modalities and Preprocessing** | |
| Modalities | Video, Audio, Text |
| Face/Human Detection | YOLOv8 Varghese & Sambath (2024) |
| Face Presence Threshold | Face detected in $\geq 90\%$ of frames |
| **Annotation and Labeling Strategy** | |
| Annotation Models | DistilRoBERTa (Text) Sanh et al. (2019), Wav2Vec 2.0 (Audio) Baevski et al. (2020) |
| Label Type | Single dominant emotion per clip |
| Label Fusion | Logit-Based Fusion from Text and Audio modalities |
| **Dataset Spliting Strategy** | |
| Movie/franchise-level | All clips from the same movie, sequel, or multi-episode series are assigned exclusively to one split (train, validation, or test) to ensure a fully unseen test set. |
| **Emotion Classes** | |
| Categories | Anger, Disgust, Fear, Joy, Sadness, Surprise, Neutral |

## 3.2 ANNOTATION METHODOLOGY

We adopted seven discrete emotion classes to ensure comparability with established benchmarks, aligning with Ekman's basic emotions framework Ekman (1992). We used domain-specific pre-trained models to annotate each modality. To avoid any conflicts between the labels, we selected models that use the same set of emotion classes across text and audio. These classes include: **Anger**, **Disgust**, **Fear**, **Joy**, **Sadness**, **Surprise**, and **Neutral**. This ensures that the annotations are consistent when combining information from all three modalities. $E$ = [Anger, Disgust, Fear, Joy, Sadness, Surprise, Neutral]

**Text Annotation:** We apply a fine-tuned **DistilRoBERTa** Sanh et al. (2019) model for emotional content analysis of dialogue text. For each utterance, we obtain a vector of real-valued scores called sentiment logits, representing the unnormalized model confidence for each emotion class in $E$.

$$l_k^{\text{text}} = \text{logits}_{\text{text}}(T_k) \in \mathbb{R}^{|E|} \tag{3}$$

**Audio Annotation:** Similarly, the acoustic segment $A_k$ is analyzed using a **wav2vec 2.0** Baevski et al. (2020) pretrained model:

$$l_k^{\text{audio}} = \text{logits}_{\text{audio}}(A_k) \in \mathbb{R}^{|E|} \tag{4}$$

**Sentiment Logits for Fusion:**  The sentiment logits $l_k^{\text{text}}$ and $l_k^{\text{audio}}$ serve as input to our logit-based fusion mechanism. Rather than relying solely on the top-class prediction, we use the full logit distributions to capture detailed emotion signals and uncertainty across modalities. The detailed explanation is given below.

**Logit-Based Multimodal Fusion for Supervised Emotion Labeling:** To annotate emotions without relying on the noisy visual modality, we propose a logit-based fusion strategy using pre-trained emotion classifiers for Text and Audio. For a fixed emotion label set $E = \{e_1, \ldots, e_7\}$, each clip produces two 7-dimensional logit vectors: $L_t = [l_{t,1}, \ldots, l_{t,7}]$ from the text model and $L_a = [l_{a,1}, \ldots, l_{a,7}]$ from the audio model. These are unnormalized scores, i.e., $l_{m,i} \in \mathbb{R}$ where $m \in \{t, a\}$.

Assuming a uniform prior $P(e_i) = 1/|E|$, the posterior over emotion class $e_i$ is modeled as:

$$P(e_i|L_t, L_a) \propto P(L_t|e_i) \cdot P(L_a|e_i) \tag{5}$$

where likelihoods are softmax-normalized:

$$\tilde{P}_m(e_i) = \frac{\exp(l_{m,i})}{\sum_{j=1}^{7} \exp(l_{m,j})}, \quad m \in \{t, a\} \tag{6}$$

To encourage modality agreement, we introduce a KL-divergence penalty:

$$D_{\text{KL}}(\tilde{P}_t || \tilde{P}_a) = \sum_{i=1}^{7} \tilde{P}_t(e_i) \log \left( \frac{\tilde{P}_t(e_i)}{\tilde{P}_a(e_i)} \right) \tag{7}$$

This yields a fused decision score:

$$S(e_i) = \log \tilde{P}_t(e_i) + \log \tilde{P}_a(e_i) - \lambda D_{\text{KL}}(\tilde{P}_t || \tilde{P}_a) \tag{8}$$

where $\lambda = 0.5$. The final emotion label is:

$$e^* = \arg \max_{e_i \in E} S(e_i) \tag{9}$$

and its confidence is normalized using:

$$F(e^*) = \frac{1}{1 + \exp(-S(e^*))}, \quad F(e^*) \in [0, 1] \tag{10}$$

This formulation captures full distributional uncertainty from both modalities and enforces semantic coherence between textual and acoustic cues. Unlike majority voting or hard max fusion, the KL divergence regularizer penalizes disagreements and rewards confident, aligned predictions.

For example, if $L_t = [0.02, \dots, 0.75]$ and $L_a = [0.05, \dots, 0.75]$ both peak at "surprise," we obtain $S(\text{surprise}) \approx -2.60$ and $F \approx 0.07$, compared to $S \approx -3.18$ and $F \approx 0.04$ in cases of disagreement. Thus, the framework promotes high-confidence, consistent labeling across modalities.

We further use the agreement condition $\arg \max L_t = \arg \max L_a$ and confidence score $F(e^*)$ to filter noisy labels. As both logit vectors come from pretrained emotion models (DistilRoBERTa and Wav2Vec 2.0), the approach scales efficiently across large unlabeled datasets and acts as a pseudo-supervisor for high-quality emotion annotation. The overall dataset annotation pipeline is summarized in Algorithm 1, which details the multimodal annotation procedure. Figure 3 illustrates the construction pipeline of the SpEmoC dataset, outlining the key steps involved in its development. Further information on annotaion file is detailed in the **Appendix**B

---

**Algorithm 1** Multimodal Dataset Annotation Procedure

**Input:** Video clips $V = \{v_1, v_2, \dots, v_N\}$ with synchronized text $T$, audio $A$, and visual frames $F$
**Output:** Annotated dataset $\mathcal{D}$ with final emotion labels $e_i^*$ for each clip
**Step 1: Preprocessing** Initialize $\mathcal{D} \leftarrow \emptyset$ **foreach** *clip* $v_i \in V$ **do**
    Extract text $T_i$, audio $A_i$, and visual frames $F_i$ from $v_i$ Detect face and human bounding boxes using
    YOLOv8:    $(x_f, y_f, w_f, h_f)$ and $(x_h, y_h, w_h, h_h)$
**end**
**Step 2: Annotation of Modalities** **foreach** *clip* $(T_i, A_i, F_i)$ **do**
    Compute text emotion logits $l_i^{\text{text}}$ using DistillRoBERTa:    $l_i^{\text{text}} = \text{logits}_{\text{text}}(T_i) \in \mathbb{R}^{|E|}$ Compute audio
    emotion logits $l_i^{\text{audio}}$ using Wav2Vec:    $l_i^{\text{audio}} = \text{logits}_{\text{audio}}(A_i) \in \mathbb{R}^{|E|}$
**end**
**Step 3: Multimodal Fusion** Fuse logits across modalities to compute final emotion score:    $S(e_i) = \log P_i(e_i) + \lambda D_{\text{KL}}(P_i^{\text{text}} || P_i^{\text{audio}})$ where $\lambda = 0.5$ Assign final label:    $e_i^* = \arg \max S(e_i)$
**Step 4: Dataset Construction** Construct annotated dataset entry:    $\mathcal{D} \leftarrow \mathcal{D} \cup \{(T_i, A_i, F_i, e_i^*)\}$
**return** $\mathcal{D}$

---

### 3.3 DATASET REFINEMENT

The initial 306,544 annotated clips were analyzed for class distribution, revealing a significant imbalance, with the neutral class dominating, as shown in Figure 4. This imbalance could bias model training, as neutral emotion appeared more frequently in the dataset. To address this, we applied threshold-based filtering to reduce the number of neutral clips, resulting in a refined dataset of 50,000 clips. After this filtering, we perform the human validation of labels. Thereafter, we obtained 30,000 coarse-grained refined clips (which is still relatively high as compared to the existing datasets) with a more balanced distribution across the seven emotion categories as shown in Figure 1 (a). The detailed analysis of this refinement process is further elaborated in **Appendix** C.

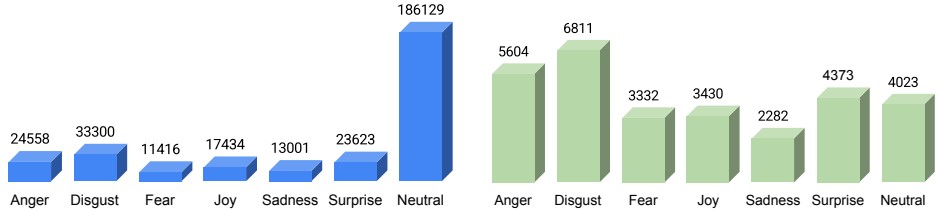

Figure 3: Overview of the SpEmoC dataset construction pipeline. Raw videos are processed to extract synchronized text ($T_k$), audio ($A_k$), and visual clips ($v_k$). Human and face bounding boxes are detected using YOLOv8: Human Box $= (x_h, y_h, w_h, h_h)$, Face Box $= (x_f, y_f, w_f, h_f)$. Emotion logits $l_k^{\text{text}}$ and $l_k^{\text{audio}}$ are computed using pretrained classifiers and fused to produce the final emotion label $e_i^*$. This process is applied across all $N_{\text{clips}}$ to construct the dataset.

Figure 4: Emotion class distribution before (see right plot in blue) and after filtering (left plot in green). The initial 306,544 clips are heavily dominated by the *neutral* class (over 186,000 samples), with underrepresentation in *fear*. A two-step filtering process was applied: (i) threshold-based filtering, retaining clips with face presence in at least 90% of frames to enhance emotional salience, and (ii) human annotation validation to remove ambiguous cases and confirm label reliability. The refined 30,000-clip dataset shows a more balanced distribution across all seven emotion classes.

**Human Validation of Labels**   To validate the reliability of SpEmoC, we conducted a human annotation study on 50,000 clips, detailed explanation provided in the **Appendix** C.1

**Dataset Splitting Strategy:**   To ensure realistic evaluation and prevent content leakage, we adopt a movie-level splitting strategy, assigning entire movies exclusively to training (70%), validation (10%), or test (20%) sets, ensuring no overlap of scenes, characters, or dialogue contexts. This movie-independent approach, applied even to franchise sequels or multi-episode series, places all related episodes in one split, guaranteeing an unseen test set free of recurring patterns or actors for robust real-world generalization. The 30,000 refined clips are distributed accordingly, with robustness enhanced by this strategy, while emotion class distributions are compared with MELD and CAER in Figure 1 (a) and Table 3 detailing class distributions across splits. Refer to **Appendix** D for details on clip distribution across splits and **Appendix** E for dataset demographic .

Table 3: Comparison of emotion class distribution between the MELD dataset Poria et al. (2018), the CAER dataset Lee et al. (2019), and the proposed SpEmoC dataset across training, development, and test splits. Red-highlighted indicates underrepresented classes (e.g., Disgust, Fear) with low sample counts, while Blue-highlighted denotes the overrepresented Neutral class with the highest sample counts in the existing datasets.

| Categories | MELD | | | CAER | | | SpEmoC (Ours) | | |
|---|---|---|---|---|---|---|---|---|---|
| | Train | Val | Test | Train | Val | Test | Train | Val | Test |
| Anger | 1109 | 153 | 345 | 1136 | 162 | 325 | 3980 | 392 | 1271 |
| Disgust | 271 | 22 | 68 | 500 | 71 | 145 | 4946 | 551 | 1298 |
| Fear | 268 | 40 | 50 | 358 | 51 | 102 | 2378 | 226 | 729 |
| Joy | 1743 | 163 | 402 | 1905 | 272 | 544 | 2506 | 340 | 578 |
| Neutral | 4710 | 470 | 1256 | 3202 | 457 | 915 | 2804 | 320 | 895 |
| Sadness | 683 | 111 | 208 | 1028 | 146 | 294 | 1612 | 195 | 474 |
| Surprise | 1205 | 150 | 281 | 1093 | 157 | 315 | 3181 | 376 | 811 |

### 3.4 SUMMARY AND FUTURE DIRECTIONS

We presented **SpEmoC**, a large-scale multimodal emotion recognition dataset comprising 306,544 clips from 3,100 English-language movies and TV series. Each clip includes synchronized video, audio, and text, focusing on speaking segments with at least 12 words and terminal punctuation to ensure emotional richness. Using pretrained models DistilRoBERTa (text), Wav2Vec 2.0 (audio), and YOLOv8 (visual) we automatically annotated and filtered the data to a high-quality, class balanced subset of 30,000 clips. SpEmoC emphasizes modality alignment and authenticity by excluding dubbed or subtitled content and includes diverse real-world conditions (e.g., grayscale, low-light, variable resolution). It supports robust learning from integrated visual, auditory, and linguistic signals. In the future, we plan to add more emotion classes, include continuous labels such as valence–arousal, and give more focus to real, non-acted samples to make the dataset more authentic and useful.

**Ethical Considerations:** We prioritize copyright and responsible use in constructing SpEmoC. The dataset is derived from publicly available movies and TV series and will be released strictly under fair-use provisions for non-commercial research. Distribution will be governed by an End User License Agreement (EULA), requiring researchers to apply for access and comply with clearly defined terms. To ensure transparency, the dataset repository will provide detailed documentation on usage boundaries, licensing conditions, and ethical safeguards.

**Dataset and code link :** The dataset (test set for evaluation) and code are available in the anonymous link provided here : `https://github.com/emouser2023/emodata.git`

## 4 BASELINE MODEL

Our baseline model integrates video, text, and audio modalities using pretrained encoders followed by a lightweight fusion classifier as shown in Figure 5.

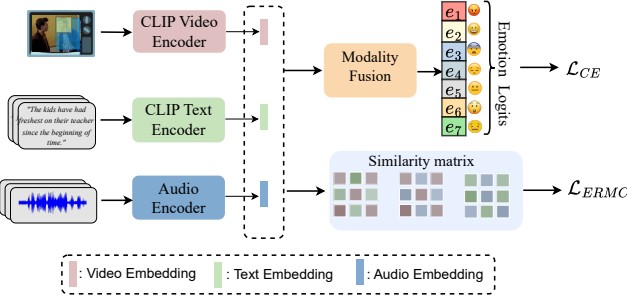

Figure 5: Illustration of the proposed multimodal emotion recognition framework. Video, text, and audio inputs are encoded with modality-specific encoders, while face and body bounding boxes provide subject-focused attention. Embeddings are fused to produce emotion logits optimized with cross-entropy loss ($\mathcal{L}_{\text{CE}}$). In parallel Extended Reweighted Multimodal Contrastive Loss ($\mathcal{L}_{\text{ERMC}}$) aligns cross-modal embeddings for robust recognition.

The video encoder uses CLIP-ViT with temporal adaptation via AIM Yang et al. (2023); the text encoder is adapted with T2L Ahmad et al.. Audio features are extracted using a pretrained HuBERT model Hsu et al. (2021). Let $\mathbf{v}, \mathbf{t}, \mathbf{a} \in \mathbb{R}^d$ be the modality embeddings. These are concatenated and passed through a two-layer MLP:

$$\mathbf{z} = [\mathbf{v}\|\mathbf{t}\|\mathbf{a}], \quad \mathbf{y} = W_2 \, \text{ReLU}(W_1\mathbf{z} + \mathbf{b}_1) + \mathbf{b}_2 \tag{11}$$

yielding logits $\mathbf{y} \in \mathbb{R}^K$ over emotion classes. Full encoder adaptation equations (AIM and T2L) are provided in **Appendix** H.

### 4.1 PROPOSED EXTENDED REWEIGHTED MULTIMODAL CONTRASTIVE LOSS (ERMC)

To align video, audio, and text embeddings semantically and emotionally, we propose an **Extended Reweighted Multimodal Contrastive (ERMC) Loss**. It computes cosine similarities across modality pairs and adjusts them using sentiment-based reweighting derived from unimodal classifiers.

**Similarity Scores:** For a batch of $N$ samples, we compute scaled cosine similarity between all pairs of modalities using a learnable temperature parameter $\tau$:

$$\mathbf{L}_{vt}^{(i,j)} = \frac{1}{\tau}\langle \mathbf{v}_i, \mathbf{t}_j \rangle, \quad \mathbf{L}_{va}^{(i,j)} = \frac{1}{\tau}\langle \mathbf{v}_i, \mathbf{a}_j \rangle, \quad \mathbf{L}_{ta}^{(i,j)} = \frac{1}{\tau}\langle \mathbf{t}_i, \mathbf{a}_j \rangle \tag{12}$$

**Sentiment-Based Reweighting:** We compute reweighting factors based on the Kullback-Leibler (KL) divergence between sentiment distributions:

$$w_{ij}^{(t)} = \frac{1}{\mathrm{KL}(\mathbf{s}_i^{(t)} \parallel \mathbf{s}_j^{(t)}) + \epsilon}, \quad w_{ij}^{(a)} = \frac{1}{\mathrm{KL}(\mathbf{s}_i^{(a)} \parallel \mathbf{s}_j^{(a)}) + \epsilon}, \quad w_{ij}^{(ta)} = \frac{1}{\mathrm{KL}(\mathbf{s}_i^{(t)} \parallel \mathbf{s}_j^{(a)}) + \epsilon} \tag{13}$$

where $\epsilon$ is a small constant for numerical stability.

**Adjusted Similarity Logits:** The reweighted similarity logits are adjusted as follows:

$$\tilde{\mathbf{L}}_{vt}^{(i,j)} = \mathbf{L}_{vt}^{(i,j)} - \lambda w_{ij}^{(t)}, \quad \tilde{\mathbf{L}}_{va}^{(i,j)} = \mathbf{L}_{va}^{(i,j)} - \lambda w_{ij}^{(a)}, \quad \tilde{\mathbf{L}}_{ta}^{(i,j)} = \mathbf{L}_{ta}^{(i,j)} - \lambda w_{ij}^{(ta)} \tag{14}$$

Here, $\lambda$ is a hyperparameter that controls the effect of sentiment reweighting.

**Contrastive Loss:** For each modality pair $(x,y) \in \{(v,t),(v,a),(t,a)\}$, we define the standard cross-entropy contrastive loss:

$$\mathcal{L}_{xy} = \frac{1}{N}\sum_{i=1}^{N} -\log \frac{\exp(\tilde{\mathbf{L}}_{xy}^{(i,i)})}{\sum_{j=1}^{N}\exp(\tilde{\mathbf{L}}_{xy}^{(i,j)})} \tag{15}$$

**Final Objective:** The complete ERMC loss is the average of all six symmetric modality pair losses:

$$\mathcal{L}_{ERMC} = \frac{1}{6}\left(\mathcal{L}_{vt} + \mathcal{L}_{tv} + \mathcal{L}_{va} + \mathcal{L}_{av} + \mathcal{L}_{ta} + \mathcal{L}_{at}\right) \tag{16}$$

The final training objective is:

$$\mathcal{L}_{\text{total}} = \mathcal{L}_{\text{CE}} + \mathcal{L}_{\text{ERMC}} \tag{17}$$

where $\mathcal{L}_{\text{CE}}$ is standard cross-entropy, and $\mathcal{L}_{\text{ERMC}}$ ensures modality consistency.

## 5 EXPERIMENTS AND RESULTS

In Table 4, we compare per-class recognition performance across MELD, CAER, and SpEmoC, where MELD and CAER show high scores for *Neutral* and *Joy* but poor results for minority classes like *Fear* (0.00 in MELD, 13.58 in CAER) and *Disgust* (2.90 in MELD, 12.24 in CAER). SpEmoC achieves consistent F1-scores above 64% for *Sadness*, *Joy*, *Disgust*, and *Anger*, with significant improvements for *Fear* (68.84) and *Disgust* (67.13), reflecting its effective class balance. With the highest overall weighted F1-score (67.84) compared to MELD (57.61) and CAER (44.04), SpEmoC proves a robust benchmark. In addition, the comparision of State-of-the-Art Methods on MELD and the stronger-baseline experiments are provided in **Appendix** F.2 and **Appendix** F.1 .

Table 4: Per-class emotion recognition performance (F1-scores) on MELD Poria et al. (2018), CAERLee et al. (2019), and the proposed **SpEmoC** dataset. SpEmoC achieves more balanced performance across all emotion categories, particularly improving underrepresented classes such as *Fear*, *Disgust*, and *Anger*, while also yielding the highest weighted F1 (W-F1) score. An upward arrow ($\uparrow$) signifies that higher values are better.

| Datasets | Neutral | Surprise | Fear | Sadness | Joy | Disgust | Anger | W-F1 $\uparrow$ |
|---|---|---|---|---|---|---|---|---|
| MELD | **76.37** | 52.05 | 0.00 | 20.77 | 55.27 | 2.90 | 38.06 | 57.61 |
| CAER | 57.01 | 32.58 | 13.58 | 27.85 | 60.20 | 12.24 | 29.33 | 44.04 |
| **SpEmoC (Ours)** | 53.11 | **76.51** | **68.84** | **64.56** | **82.62** | **67.13** | **67.28** | **67.84** |

**Cross-Dataset (Out-of-Domain) Evaluation:** We have conducted cross-dataset (Train $\rightarrow$ Test ) evaluations. We have trained a multimodal model (TLC-MAP) Zhou et al. (2024) solely on SpEmoC and directly tested it on MELD and CAER without any fine-tuning. This evaluates out-of-domain robustness, and the results are summarized in Table 5. We observe that models trained on SpEmoC generalize better to MELD and CAER than models trained directly on those datasets. In contrast, MELD $\rightarrow$ SpEmoC and CAER $\rightarrow$ SpEmoC transfers show substantial performance drops, indicating limited representational richness in MELD and CAER. Meanwhile, SpEmoC $\rightarrow$ MELD and SpEmoC $\rightarrow$ CAER maintain moderate and stable performance, demonstrating that SpEmoC supports upward transfer to smaller benchmarks. Finally, SpEmoC $\rightarrow$ SpEmoC achieves strong in-domain performance, reflecting the dataset's internal consistency and high-quality annotations. Additional cross-dataset results for the proposed model are provided in **Appendix** F.3 (Table 17).

Table 5: Cross-dataset generalization across SpEmoC (ours), MELD, and CAER using the TLC-MAP Zhou et al. (2024) model.

| Train → Test | Surprise | Joy | Fear | Disgust | Anger | Neutral | Sadness | W-F1 ↑ | Δ Gain |
|---|---|---|---|---|---|---|---|---|---|
| MELD → MELD | 56.04 | 56.44 | 17.07 | 22.22 | 46.05 | 77.75 | 33.93 | 62.68 | -11.95 |
| SpEmoC → MELD | 45.19 | 29.37 | 3.96 | 16.72 | 39.71 | 69.59 | 25.24 | 50.73 | |
| SpEmoC → SpEmoC | 75.51 | 74.36 | 71.13 | 72.25 | 75.55 | 70.84 | 73.76 | 73.67 | -37.67 |
| MELD → SpEmoC | 44.83 | 53.47 | 3.92 | 18.21 | 51.28 | 34.91 | 30.77 | 36.00 | |
| CAER → CAER | 13.24 | 28.73 | 10.85 | 7.51 | 23.37 | 44.79 | 12.98 | 28.26 | +2.49 |
| SpEmoC → CAER | 21.31 | 17.92 | 10.78 | 13.30 | 20.30 | 46.69 | 11.36 | 30.75 | |
| SpEmoC → SpEmoC | 75.51 | 74.36 | 71.13 | 72.25 | 75.55 | 70.84 | 73.76 | 73.67 | -49.03 |
| CAER → SpEmoC | 21.14 | 34.79 | 3.49 | 5.47 | 40.94 | 25.89 | 17.07 | 24.64 | |
| MELD → MELD | 56.04 | 56.44 | 17.07 | 22.22 | 46.05 | 77.75 | 33.93 | 62.68 | -28.06 |
| CAER → MELD | 19.46 | 17.44 | 5.13 | 1.14 | 16.09 | 50.71 | 13.56 | 34.62 | |
| CAER → CAER | 13.24 | 28.73 | 10.85 | 7.51 | 23.37 | 44.79 | 12.98 | 28.26 | +35.95 |
| MELD → CAER | 53.27 | 57.32 | 15.58 | 17.82 | 44.59 | 79.91 | 34.06 | 64.21 | |

**Evaluation of State-of-the-Art Methods on SpEmoC:** We evaluate our baseline model against several state-of-the-art (SOTA) multimodal fusion frameworks, including MulT Tsai et al. (2019), MISA Hazarika et al. (2020), EmotionCLIP Zhang et al. (2023) and TLC-MAP Zhou et al. (2024). All models are trained and tested on the SpEmoC dataset using identical train-validation-test splits (70%/10%/20%) and optimization settings for a fair comparison. As shown in Table 6, our model outperforms existing methods across most emotion categories, with notable gains for underrepresented emotions such as *Fear* (68.84 F1) and *Disgust* (67.13 F1). Our baseline also achieves the highest weighted F1-score (67.84), outperforming MulT (53.37), MISA (50.78), and EmotionCLIP (51.30), and is surpassed only by the TLC-MAP (73.67). These findings demonstrate that SpEmoC effectively addresses class imbalance, establishing it as a reliable benchmark for multimodal emotion recognition.

Table 6: Per-class F1-scores of state-of-the-art multimodal emotion recognition methods on the proposed **SpEmoC** dataset.

| Methods | Neutral | Surprise | Fear | Sadness | Joy | Disgust | Anger | W-F1 ↑ |
|---|---|---|---|---|---|---|---|---|
| MulT Tsai et al. (2019) | 35.13 | 47.10 | 40.44 | 60.47 | 75.26 | 60.06 | 60.05 | 53.37 |
| MISA Hazarika et al. (2020) | 32.40 | 41.30 | 36.00 | 51.70 | 66.40 | 49.80 | 50.90 | 50.78 |
| EmotionCLIP Zhang et al. (2023) | 31.76 | 52.63 | 50.49 | 48.47 | 66.93 | 55.82 | 51.60 | 51.30 |
| TLC-MAP Zhou et al. (2024) | 75.51 | 74.36 | 71.13 | 72.25 | 75.55 | 70.84 | 73.76 | 73.67 |
| **SpEmoC (Ours)** | **53.11** | **76.51** | **68.84** | **64.56** | **82.62** | **67.13** | **67.28** | **67.84** |

## 5.1 ABLATION STUDY

**Low-Resource Fine-Tuning:** We conducted low-resource finetuning experiments on MELD and CAER using only 10%, 30%, and 50% of the training data (Table 7). SpEmoC-pretrained models consistently outperformed non-pretrained baselines, confirming its value for data-efficient learning. Additional unimodal and multimodal backbone improvements, along with ablations on transfer learning, loss functions, neutral-class removal, and modality analysis, are provided in Appendix G.

Table 7: SpEmoC pretraining boosts MELD and CAER performance in low-data settings. Results for MELD are highlighted in red, and CAER results are shown in green.

| Training Split | Baseline | +SpEmoC Pretraining (W-F1 ↑ ) |
|---|---|---|
| 10% | 51.13 / 25.44 | **53.23** / **26.77** |
| 30% | 53.88 / 26.69 | **56.54** / **40.66** |
| 50% | 55.14 / 43.71 | **57.60** / **44.42** |

## 6 CONCLUSION AND FUTURE WORK

We introduced **SpEmoC**, a large-scale multimodal dataset with 30,000 refined clips from 306,544 segments across 3,100 English-language movies and TV series, offering synchronized **visual**, **audio**, and **text** modalities annotated for seven emotions. Unlike existing datasets, SpEmoC is class-balanced, enabling fair learning and balanced F1-scores across all emotions, including underrepresented ones like *fear* and *disgust*. We developed an automated annotation pipeline using pretrained models (Wav2Vec, DistilRoBERTa) with human validation, and a lightweight baseline model with Extended Reweighted Multimodal Contrastive (ERMC) Loss, achieving a 67.84% F1-score with 8.68M parameters. This foundation addresses scale, modality alignment, imbalance, and efficiency, paving the way for future enhancements including non-acted real-world videos, continuous valence-arousal labels, and physiological signals.

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

# Appendix

**Dataset and Code Availability :** The dataset (test set for evaluation) and code are available in the anonymous link provided here: `https://github.com/emouser2023/emodata.git`

## A  DATA COLLECTION

The videos for the SpEmoC dataset were collected using a Python-based implementation of the YouTube API, specifically youtube-search-python Mercer (2021), which replicates the search behavior of the YouTube web interface. We used search queries such as "TV series", "movies," and "TV shows" to identify long-form content rich in emotional expression. To maintain linguistic consistency, we filtered the results to include only English-language videos without dubbing. Additionally, to exclude short or irrelevant clips and ensure meaningful emotional content, we retained only videos longer than 40 minutes. This filtering process resulted in a curated set of 3,100 videos from diverse TV shows and movies, covering a wide range of demographics, genres, and authenticity of affect, as summarized in Table 8.

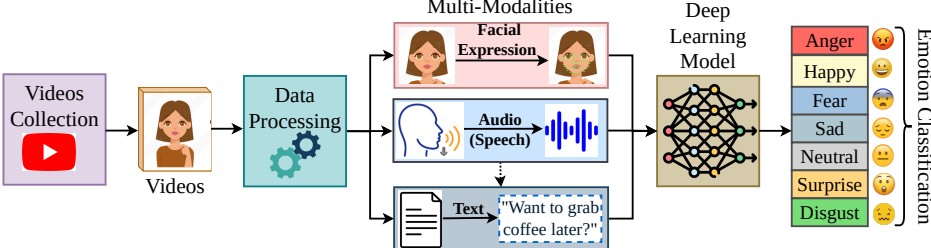

Figure 6: Overview of the multi-modal emotion recognition pipeline. Videos are collected from online sources (e.g., YouTube) and undergo preprocessing to extract three primary modalities: visual, speech audio, and textual transcripts. Each modality is analyzed individually and then fused through a deep learning model to perform emotion classification into seven categories: Anger, Happy, Fear, Sad, Neutral, Surprise, and Disgust.

Table 8: Demographic, genre, and authenticity distribution of the 3100 videos.

| Category | Distribution (approx.) | Notes |
|---|---|---|
| **Ethnicity** | Western/White: 60%, Asian: 20%, African/Black: 12%, Other: 8% | Skewed toward Western media; noted as a limitation |
| **Genres** | Drama: 30%, Comedy: 20%, Romance: 15%, Thriller: 15%, Horror: 10%, History: 10% | Wide genre diversity, reflecting emotional variation |
| **Authenticity of Affect** | Acted (Movies/TV): ∼85%, Genuine (Interviews, Documentaries, Reactions): ∼15% | Mix of acted and spontaneous expressions; genuine subset improves authenticity |

## B  ANNOTATION FILE INFORMATION

The table 9 provides an annotation summary for a 4-second clip (1230.1s to 1234.02s) from "5THE BRIEF Blame," clip 43, containing 74 frames. It includes metadata such as the text "I don't doubt that you were genuinely alarmed by what you saw," emotion scores with text logits showing a high probability for a specific emotion (0.9532) and a neutral score of 0.0141, and audio logits with a neutral score of -0.0092. Detection and fusion results indicate perfect confidence in face and human detection (1.0), a final emotion label of "Fear," inconsistency between modalities (False), and a fusion score of 0.0411. The clip meets the filtering criteria ($f^k > 0.9$, $w_t^k < 0.05$, $w_a^k < 0.05$), ensuring its suitability for multimodal emotion analysis in the SpEmoC dataset. Figure 7 visualizes the synchronization process, showing the alignment of video frames, audio, and transcripts, along with human and face bounding boxes over a representative 4-second clip from the SpEmoC dataset.

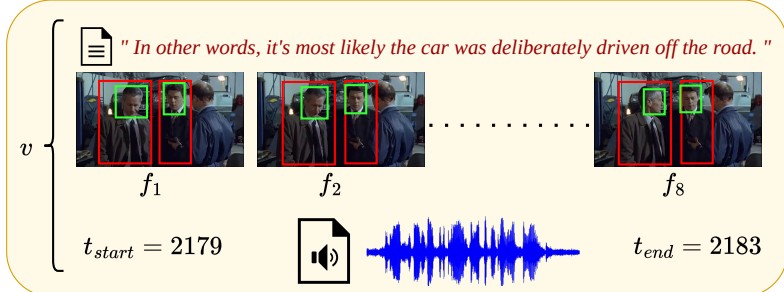

Figure 7: Synchronization of modalities for a 4-second clip $v$ from the SpEmoC dataset, with $t_{\text{start}} = 2179$ seconds and $t_{\text{end}} = 2183$ seconds. The clip includes frames $f_1 \ldots f_8$, aligned with corresponding audio and transcript text, and annotated with human and face bounding boxes to support multimodal emotion recognition.

Table 9: Annotation summary of a sample clip from 5THE BRIEF Blame, clip 43.

| **Metadata** | |
|---|---|
| Clip Identifier | 5THE BRIEF Blame, Clip 43 |
| Duration (s) | (Start: 1230.1, End: 1234.02) |
| Text | "I don't doubt that you were genuinely alarmed by what you saw." |
| Number of Frames | 74 |
| **Emotion Scores** | |
| Text Emotion Logits | [0.0062, 0.0026, 0.9532, 0.0015, 0.0141, 0.0024, 0.0200] |
| Text Neutral Score | 0.0141 |
| Audio Emotion Logits | [-0.0500, 0.0340, 0.0282, 0.0154, -0.0092, -0.0892, 0.0115] |
| Audio Neutral Score | -0.0092 |
| **Detection and Fusion** | |
| Face Detection Confidence ($f^k$) | 1.0 |
| Human Detection Confidence | 1.0 |
| Final Emotion Label | Fear |
| Is Consistent? | False |
| Fusion Score | 0.0411 |
| **Filtering Status** | |
| Filtering Criteria | $(f^k > 0.9, w_t^k < 0.05, w_a^k < 0.05)$ |

## C   DATSET FILTERING

**Filtering Process:** To obtain a refined dataset for a balanced class distribution, we curated the data using a multi-step filtering strategy. We implemented a meticulous filtering process to address the dominance of neutral clips observed in the initial 306,544 clips, focusing on evaluating neutral score thresholds for text and audio modalities. This ensures the presence of faces in visual frames to retain clips with strong emotional signals. This process was informed by manual experimentation with multiple thresholds and validated through performance analysis, as detailed below. Consequently, we observed that many clips, particularly those labeled as neutral, contained text and audio with high neutral scores, indicating weak emotional content. Whereas these scores, derived from pretrained models DistilRoBERTa for text Sanh et al. (2019) and Wav2Vec 2.0 for audio Baevski et al. (2020), represent the probability of neutrality after applying a sigmoid transformation to the logits.

For a clip $c_k$, let:

- $l_t^k = \text{text\_neutral\_logit}(T_k) \in \mathbb{R}$: the neutral logit for text,
- $l_a^k = \text{audio\_neutral\_logit}(A_k) \in \mathbb{R}$: the neutral logit for audio,
- $f^k = \text{has\_face}(v_k) \in [0,1]$: the confidence score indicating the presence of a face in the visual frames.

These logits are converted to probabilities via the sigmoid function:

$$w_t^k = \sigma(l_t^k) = \frac{1}{1 + e^{-l_t^k}}, \quad w_a^k = \sigma(l_a^k) = \frac{1}{1 + e^{-l_a^k}},$$

where $w_t^k, w_a^k \in [0,1]$ represent the probability that the text or audio expresses a neutral sentiment. To filter the dataset, we manually tested multiple neutral score thresholds for text ($\theta_t$) and audio ($\theta_a$), alongside a face detection threshold ($\theta_f$).

We retained a clip $c_k$ if the following conditions were met:

$$\text{Retain } c_k \text{ if } \begin{cases} f^k \geq \theta_f \\ w_t^k < \theta_t \text{ and } w_a^k < \theta_a, & \text{if } e_j^* \neq \text{neutral} \end{cases}$$

where $e_i^*$ is the final emotion label.

Here, $\theta_f = 0.9$ ensures that a face is detected in at least 90% of the video frames, while $\theta_t, \theta_a = 0.05$ filter out samples with weak emotional content in text and audio. For class balancing, we included a small subset of neutral clips, approximately 15% relative to the number of non-neutral clips, by relaxing these thresholds. Our filtering strategy substantially shifts the distribution of neutral probabilities toward lower values, resulting in a refined 50,000 clips with stronger emotional cues across all modalities.

### C.1 HUMAN ANNOTATION

To validate the reliability of SpEmoC, we conducted a human annotation study on this filtered subset (50,000 clips), ensuring high quality and balance. Twenty expert annotators, proficient in English and trained on standardized guidelines based on Ekman's framework Ekman (1992), were selected. Each clip was independently reviewed by at least three annotators using all modalities (text, audio, visual), and final labels were assigned via majority voting. Inter-annotator agreement reached a Fleiss' Kappa of 0.62 (substantial agreement Landis & Koch (1977)). This process eliminated ambiguous clips, yielding the final 30,000 balanced clips, as shown in Figure 4. This combined threshold-based filtering and human annotation not only mitigates class imbalance but also enhances label reliability.

### C.2 LIMITATION OF HUMAN ANNOTATION

Although human annotation provides valuable ground truth for emotion labeling in SpEmoC, it is inherently subject to limitations that can impact reliability. Annotators bring their own subjective perspectives, shaped by personal experiences, cultural backgrounds, and interpretive biases, which may lead to inconsistent classifications of the same multimodal clip. Furthermore, distinguishing between closely related emotions such as *surprise* and *joy*, *fear* and *anger*, or *disgust* and *fear* is particularly challenging due to overlapping expressive cues in facial, vocal, and textual modalities, making clear boundaries between categories difficult to establish. These ambiguities often result in misclassifications, especially in subtle or low-intensity cases. To mitigate these challenges, our pipeline integrates pretrained model predictions with human annotations, leveraging automated consistency to complement human judgment in assigning final labels, and underscoring the need for hybrid approaches in the construction of large-scale emotion datasets.

### C.3 NEUTRAL SCORE DISTRIBUTION BEFORE AND AFTER FILTERING

Figure 8 illustrates the distribution of neutral class probabilities from text and audio modalities in the full dataset (left) and the filtered 30k subset (right). The filtering process removes clips with high neutral scores, yielding a dataset with more emotionally salient and less ambiguous samples across both modalities.

## D DATASET SPLIT STRATEGY

The 30,000 refined clips are distributed as follows: 70% for training, 10% for validation (used for tuning), and 20% for testing (used for evaluating generalization to novel movies), as detailed in Table 10. This strategy enhances the robustness of performance metrics and better simulates real-world deployment by ensuring diverse representation across splits. To prevent content leakage and ensure realistic evaluation, we adopt a movie-level splitting approach, where entire movies are assigned exclusively to one of the three sets, avoiding overlap of scenes, characters, or dialogue contexts. This method is particularly effective for handling franchise sequels or multi-episode series, as all related episodes are confined to a single split, minimizing the risk of recurring visual or conversational patterns influencing model performance. This rigorous splitting strategy, combined with the balanced dataset design, supports the development of generalizable emotion recognition models.

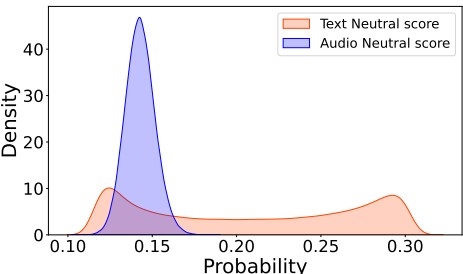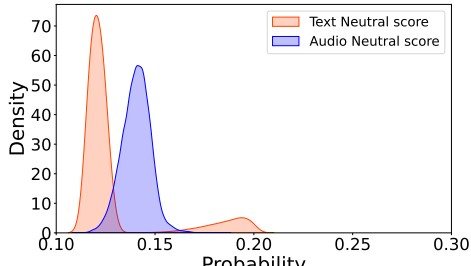

Figure 8: Distribution of neutral class probabilities from text and audio before (left) and after (right) filtering. Filtering removes emotionally ambiguous clips, shifting the distribution toward lower neutral scores and enhancing signal richness across modalities.

Table 10: Dataset Splitting Information of refined 30,000 clips

| Split | Percentage (%) | Number of Clips |
|---|---|---|
| Training Set | 70% | 21,000 |
| Validation Set | 10% | 24,00 |
| Test Set | 20% | 60,00 |
| **Total** | 100% | 30,000 |

## D.1 BALANCED EMOTION CLASS DISTRIBUTION IN SPEMOC

The class distribution of the proposed SpEmoC dataset, alongside existing datasets, is illustrated in Figure 9, which presents the percentage-wise distribution of emotion classes. This figure highlights SpEmoC's balanced representation across the seven categories (Anger: 18.8%, Disgust: 22.8%, Fear: 11.2%, Joy: 11.5%, Neutral: 13.5%, Sadness: 7.6%, Surprise: 14.6%), contrasting with the imbalanced distributions in datasets like MELD (Neutral: 47.0%) and CAER (Neutral: 34.7%), where the neutral class dominates.

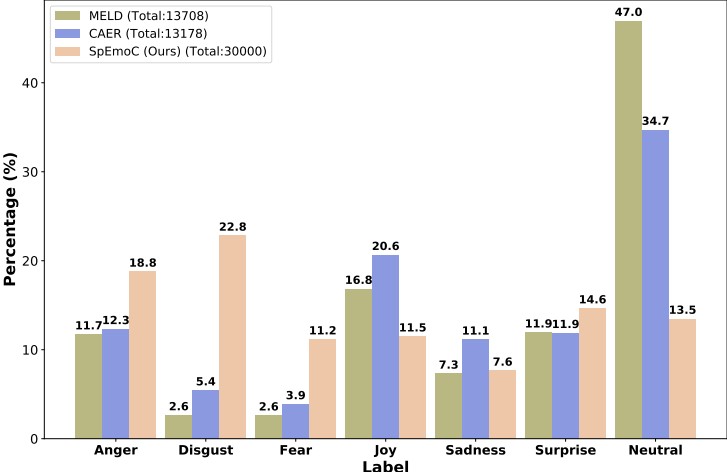

Figure 9: Comparison of emotion label distributions across MELD, CAER, and SpEmoC (Ours). While MELD and CAER exhibit strong class imbalance (e.g., Neutral dominating with 47.0% and 34.7%, respectively), SpEmoC achieves a more balanced distribution across all seven emotions, reducing bias toward majority classes.

## E SPEMOC DEMOGRAPHICS

This section presents the demographic composition of the dataset (30K refined clips) across age, gender, and ethnic groups. As shown in Tables 11, 12 and 13, the dataset includes a broad age

range from children to seniors, with substantial representation of young and middle-aged adults. The gender distribution includes both male and female participants, and the dataset spans multiple ethnic groups, providing useful diversity for analysis. These demographics give a clear overview of the dataset's population coverage.

Table 11: Age Distribution

| Age Group | Range | Count | % |
|---|---|---|---|
| Child | ≤ 12 | 235 | 0.79 |
| Teen | 13–19 | 471 | 1.58 |
| Young Adult | 20–35 | 13,366 | 44.88 |
| Adult | 36–55 | 12,169 | 40.84 |
| Senior | 56+ | 3,542 | 11.89 |

Table 12: Ethnic Group Distribution

| Ethnic Group | Count | % |
|---|---|---|
| Western | 18,500 | 62.11 |
| African | 8,300 | 27.87 |
| Asian | 1,983 | 6.66 |
| Middle Eastern | 1,000 | 3.36 |

Table 13: Gender Distribution

| Gender | Percentage |
|---|---|
| Male | 68.4% |
| Female | 31.6% |

## F    MORE EXPERIMENTS AND RESULTS

### F.1    RESULT ON STRONGER BASELINE

We evaluated a strong SOTA multimodal transformer, TCL-MAPZhou et al. (2024) (AAAI 2024), under the same preprocessing and training pipeline. TCL-MAP Zhou et al. (2024) achieves substantially higher performance across all datasets *(see Table 14 and 15)*, demonstrating that:SpEmoC supports stronger baseline models.

Table 14: Performance Comparison: Original Baseline vs. Stronger Baseline (TCL-MAP)Zhou et al. (2024).

| Dataset | Original Baseline | Stronger Baseline (TCL-MAP ?) | Improvement |
|---|---|---|---|
| MELD | 57.61 | 62.68 | **+5.07** |
| CAER | 44.04 | 28.26 | **–15.78*** |
| SpEmoC | 67.84 | 73.67 | **+5.83** |

Table 15: Stronger Baseline (TCL-MAP Zhou et al. (2024)- Per-Class Performance of MELD , CAER and SpEmoC.

| Dataset | Surprise | Joy | Fear | Disgust | Anger | Neutral | Sadness | W-F1 |
|---|---|---|---|---|---|---|---|---|
| MELD | 56.04 | 56.44 | 17.07 | 22.22 | 46.05 | 77.75 | 33.93 | 62.68 |
| CAER | 13.24 | 28.73 | 10.85 | 7.51 | 23.37 | 44.79 | 12.98 | 28.26 |
| SpEmoC | 75.51 | 74.36 | 71.13 | 72.25 | 75.55 | 70.84 | 73.76 | 73.67 |

### F.2    EVALUATION OF STATE-OF-THE-ART METHODS ON MELD

We have tabulated the performance comparison of state-of-the-art methods on the MELD dataset *(see Table 16)*. Furthermore, we deliberately exclude CAER-S Lee et al. (2019) from our comparisons. Since CAER-S contains only static images, and prior work reports results exclusively on this image-based benchmark. Since our model is explicitly designed for video-based emotion recognition and relies heavily on temporal information, comparing it directly with CAER-S would be neither meaningful nor technically consistent.

### F.3    ADDITIONAL CROSS-DATASET (OUT-OF-DOMAIN) EVALUATION

As shown in Table 17, the cross-dataset results indicate a performance drop for our baseline model. Since model design is not the primary focus of this work, we view this limitation as an opportunity to explore stronger baseline architectures in future work.

## G    ADDITIONAL ABLATION STUDY

Table 16: Performance comparison of state-of-the-art methods on the MELD dataset. Per-class F1 scores and overall weighted F1 (W-F1) are reported.

| Methods | Neutral | Surprise | Fear | Sadness | Joy | Disgust | Anger | W-F1 |
|---|---|---|---|---|---|---|---|---|
| bc-LSTM Poria et al. (2017b) | 73.8 | 47.7 | 5.4 | 25.1 | 51.32 | 5.2 | 38.4 | 55.8 |
| DialogueGCN Ghosal et al. (2019) | 72.1 | 41.7 | 2.8 | 21.1 | 44.2 | 6.7 | 36.5 | 52.85 |
| A-DMN Xing et al. (2020) | 78.9 | 55.3 | 8.6 | 24.9 | 57.4 | 3.4 | 40.9 | 60.4 |
| RGAT Ishiwatari et al. (2020) | 78.1 | 41.5 | 2.4 | 30.7 | 58.6 | 2.2 | 44.6 | 59.6 |
| CTNet Lian et al. (2021) | 77.4 | 50.3 | 10.0 | 32.5 | 56.0 | 11.2 | 44.6 | 60.2 |
| MMGCN Hu et al. (2021) | 77.1 | 53.9 | 0.0 | 17.7 | 56.9 | 0.0 | 42.6 | 59.4 |
| **TCL-MAPZhou et al. (2024) on MELD** | 77.75 | 56.04 | 17.07 | 33.93 | 56.44 | 22.22 | 46.05 | 62.68 |
| **Ours** | 76.3 | 52.0 | 0.0 | 20.7 | 55.2 | 2.9 | 38.0 | 57.6 |

Table 17: Cross-dataset across SpEmoC (ours), MELD, and CAER on proposed model.

| Train → Test | Surprise | Joy | Fear | Disgust | Anger | Neutral | Sadness | W-F1 |
|---|---|---|---|---|---|---|---|---|
| SpEmoC → MELD | 40.84 | 7.49 | 2.60 | 1.65 | 18.58 | 3.43 | 7.20 | 10.33 |
| SpEmoC → CAER | 17.25 | 8.01 | 3.29 | 5.13 | 19.82 | 3.93 | 12.94 | 9.31 |
| MELD → SpEmoC | 10.20 | 0.00 | 21.08 | 7.94 | 3.93 | 2.35 | 0.00 | 6.79 |

**Transfer Learning to External Datasets:** To evaluate cross-dataset generalisation, we pre-trained the same multimodal encoder used in baseline on SpEmoC and then fine-tuned it on two widely used emotion benchmarks: MELD and CAER. In both cases, SpEmoC pretraining led to higher weighted-F1 scores, as shown in Table 18. All experiments were conducted with a batch size of 35 and trained for 20 epochs. Per-class improvements are also consistent , indicating that SpEmoC helps representations capture fine-grained affective cues are provided in in Table 20. In contrast, as shown in Table 19, pretraining on MELD does not improve SpEmoC , suggesting that SpEmoC is a richer corpus for representation learning.

Table 18: Performance gains on MELD and CAER obtained through SpEmoC pretraining.

| Dataset | Baseline | +SpEmoC Pretraining (W-F1 ↑) | Δ (gain) |
|---|---|---|---|
| MELD | 57.6% | **60.0%** | +2.4 |
| CAER | 44.0% | **47.28%** | +3.28 |

Table 19: Performance comparison on SpEmoC with MELD-pretrained finetuning.

| Dataset | Baseline | +MELD Pretraining (W-F1 ↑) | Δ (gain) |
|---|---|---|---|
| SpEmoC | **67.84%** | 65.13% | -2.71 |

Table 20: Class-wise and weighted F1 improvements on MELD and CAER after finetuning with SpEmoC-pretrained weights.

| Dataset | Neutral | Surprise | Fear | Sadness | Joy | Disgust | Anger | W-F1 |
|---|---|---|---|---|---|---|---|---|
| **MELD Training** | 76.37 | 52.05 | 0.00 | 20.77 | 55.27 | 2.90 | 38.06 | 57.61 |
| **MELD Finetuning (SpEmoC Pretraining)** | 77.66 | 53.95 | 11.11 | 22.73 | 55.45 | 16.09 | 43.79 | 60.00 |
| **CAER Training** | 57.01 | 32.58 | 13.58 | 27.85 | 60.20 | 12.24 | 29.33 | 44.04 |
| **CAER Finetuning (SpEmoC Pretraining)** | 60.04 | 13.95 | 8.76 | 37.04 | 72.48 | 12.93 | 36.42 | 47.28 |

**Unimodal / Multimodal Backbone Improvement:** To further validate the representational benefits of SpEmoC, we evaluate both unimodal (text-only, audio-only) and multimodal (text + audio) backbones on MELD after pretraining on SpEmoC. Specifically, we first train the text-only encoder on SpEmoC and then fine-tune this pretrained text model on MELD; the same pretraining–finetuning procedure is applied to the audio-only and the combined text–audio configurations. As shown in Table 21, SpEmoC pretraining provides consistent gains across all settings, improving text-only, audio-only, and multimodal models alike. These results demonstrate that SpEmoC strengthens both modality-specific encoders and joint multimodal representations.

**Ablation On Neutral Class Removal:** Table 22 reports the performance of MELD, CAER, and SpEmoC after excluding the *Neutral* class. In MELD and CAER show modest increases in certain minority categories (e.g., *Sadness* rises from 20.77 to 49.10 in MELD, and *Joy* rises from 60.20

to 74.37 in CAER), but their performance remains inconsistent and unbalanced across categories, highlighting that their reported gains are largely inflated by the presence of Neutral. By contrast, SpEmoC achieves consistently strong recognition across all categories, with substantial gains in *Fear* (66.56), *Disgust* (68.92), and *Anger* (67.97). This demonstrates that SpEmoC does not rely on the Neutral class for performance gains and instead provides a balanced benchmark for evaluating non-neutral emotional states, reflected in the highest W-F1 score (71.03), as illustrated in Fig. 11.

Table 21: MELD performance gains from SpEmoC pretraining across unimodal (T: Text, A: Audio) and multimodal inputs.

| Dataset | Model Init | W-F1 | Gain |
|---|---|---|---|
| MELD (T) | baseline | 47.1 | — |
| MELD (T) | baseline + Pretrained SpEmoC | **49.3** | +2.2 |
| MELD (A) | baseline | 44.8 | — |
| MELD (A) | baseline + Pretrained SpEmoC | **46.1** | +1.3 |
| MELD (T + A) | baseline | 55.7 | — |
| MELD (T + A) | baseline + Pretrained SpEmoC | **57.4** | +1.7 |

Table 22: Ablation study Neutral class removal: Per-class F1-scores on MELD, CAER, and SpEmoC (ours) after removing the dominant *Neutral* class. This analysis highlights how SpEmoC achieves balanced improvements across all remaining emotions, resulting in the highest weighted F1 (W-F1).

| Datasets | Surprise | Fear | Sadness | Joy | Disgust | Anger | W-F1 ↑ |
|---|---|---|---|---|---|---|---|
| MELD | 56.95 | 0.00 | 49.10 | 66.20 | 0.00 | 45.10 | 50.38 |
| CAER | 39.87 | 27.18 | 35.16 | 74.37 | 20.66 | 42.88 | 48.19 |
| **SpEmoC (Ours)** | **74.99** | **66.56** | **66.96** | **85.62** | **68.92** | **67.97** | **71.03** |

Figure 10 effectively demonstrates the strength of our proposed model, showcasing its ability to distinctly separate all emotion classes within the embedding space, highlighting the strength of its multimodal fusion approach. Furthermore, the removal of the neutral class enhances the model's performance, enabling it to learn all emotional classes (Anger, Disgust, Fear, Joy, Sadness, Surprise) effectively without bias toward the previously dominant neutral class, ensuring robust and balanced emotion recognition.

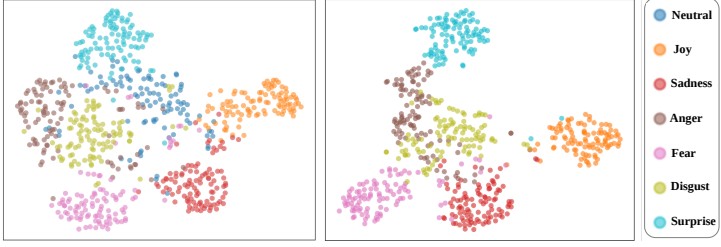

Figure 10: 2-Dimensional t-SNE visualization of feature embeddings from multimodal fusion on the SpEmoC dataset. The left plot shows clustering with all seven classes (including Neutral), while the right plot presents the distribution without the Neutral class, highlighting improved separation of minority emotions. These visualizations demonstrate clear class-wise boundaries and validate the effectiveness of the proposed model.

**Modality Ablation Study:**  Table 23 reports results across individual modalities (Text, Video, Audio) and their combinations. Single-modality performance is moderate, with text (T) performing best among unimodal inputs (W-F1 = 56.12). Pairwise fusion (T+V, T+A, V+A) consistently improves recognition, with text-based combinations yielding stronger results. The full fusion of all three modalities (T+V+A) achieves the highest per-class F1-scores and overall weighted F1 (67.84), confirming the complementary role of multimodal signals in emotion recognition.

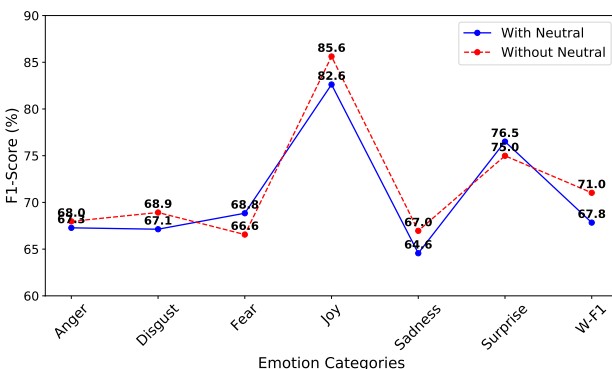

Figure 11: Comparison of F1-Scores across emotion categories with and without the Neutral class. SpEmoC achieves consistently strong recognition across all categories. This shows that SpEmoC does not depend on the Neutral class for performance gains, instead offering a balanced benchmark for non-neutral emotions. The highest weighted F1-score is observed without Neutral (71.0).

Table 23: Modality ablation study on the SpEmoC dataset, reporting per-class F1 scores, overall weighted F1 score (W-F1). SpEmoC (T+A+V) outperforms all unimodal and bimodal configurations, with significant improvements in underrepresented classes such as Fear and Disgust. Bold values indicate the best performance in each column.

| Modality | Neutral | Surprise | Fear | Sadness | Joy | Disgust | Anger | W-F1↑ |
|---|---|---|---|---|---|---|---|---|
| T | 45.32 | 62.47 | 51.16 | 48.25 | 55.38 | 49.82 | 58.67 | 56.12 |
| V | 41.76 | 59.28 | 48.39 | 45.73 | 50.24 | 47.15 | 54.92 | 53.48 |
| A | 39.84 | 57.13 | 46.75 | 44.62 | 48.91 | 45.08 | 53.26 | 52.12 |
| T+V | 48.91 | 67.52 | 55.27 | 50.83 | 57.61 | 52.14 | 61.38 | 60.47 |
| T+A | 50.37 | 69.14 | 56.93 | 52.12 | 58.48 | 53.26 | 62.87 | 62.04 |
| V+A | 47.28 | 65.87 | 54.16 | 49.97 | 56.24 | 51.03 | 60.12 | 59.72 |
| **T+V+A** | **53.11** | **76.51** | **68.84** | **64.56** | **82.62** | **67.13** | **67.28** | **67.84** |

**Impact of the Extended Reweighted Multimodal Contrastive Loss:** We evaluate the effect of our Extended Reweighted Multimodal Contrastive (ERMC) loss by comparing it against widely used alternatives, including cross-entropy, weighted cross-entropy, focal loss Lin et al. (2017), and class-balanced loss Cui et al. (2019). As shown in Table 24, incorporating ERMC alongside the standard cross-entropy improves overall F1-score, validating the benefit of sentiment-guided embedding alignment.

Table 24: Performance comparison of different loss functions.

| Loss Function | W-F1 ↑ |
|---|---|
| Cross Entropy | 65.80 |
| Weighted Cross Entropy | 66.42 |
| Focal Loss Lin et al. (2017) | 66.10 |
| Class-Balanced Loss Cui et al. (2019) | 66.70 |
| **ERMC (Ours) + CE** | **67.84** |

## H DETAILED ENCODER ADAPTATIONS

### H.1 VIDEO ENCODER VIA AIM

The video encoder in Figure Radford et al. (2021) adapts a pre-trained ViT-B/16 (CLIP) backbone for video understanding, following the AIM framework Yang et al. (2023). It processes a video clip by sampling frames at a fixed resolution ($H \times W \times C$). Each frame is split into $N = (H \times W)/P^2$ patches (with patch size $P$), mapped to $D$-dimensional embeddings, yielding $\mathbf{x}p \in \mathbb{R}^{T \times N \times D}$. A [class] token is prepended per frame, and positional embeddings $\mathbf{E}pos \in \mathbb{R}^{(N+1) \times D}$ are added, resulting in $\mathbf{z}_0 \in \mathbb{R}^{T \times (N+1) \times D}$. This input is fed into a series of transformer blocks, modified for spatiotemporal reasoning while keeping the ViT backbone frozen.

The AIM mechanism Yang et al. (2023) introduces lightweight spatial and temporal adapters within CLIP-ViT layers Radford et al. (2021),which introduces lightweight adapters into the transformer as shown in Fig. 12 (a). Spatial adaptation adds an adapter after the self-attention (S-MSA) layer in each transformer block, using a bottleneck structure to fine-tune spatial features, producing $\mathbf{z}_l^S \in \mathbb{R}^{T \times (N+1) \times D}$.

For temporal adaptation, the pre-trained self-attention layer is reused as T-MSA to model temporal relationships across frames. The input $\mathbf{z} \in \mathbb{R}^{T \times (N+1) \times D}$ is reshaped to $\mathbf{z}^T \in \mathbb{R}^{(N+1) \times T \times D}$, enabling T-MSA to capture dependencies among the $T$ frames. A temporal adapter is appended to adapt temporal features, yielding $\mathbf{z}_l^T \in \mathbb{R}^{T \times (N+1) \times D}$.

Joint adaptation adds an adapter parallel to the MLP layer, scaled by a factor $s$, for spatiotemporal tuning, resulting in $\mathbf{z}_l \in \mathbb{R}^{T \times (N+1) \times D}$. Only adapters are updated during training, and the final video representation is obtained by averaging [class] tokens across frames, producing an embedding $\in \mathbb{R}^D$ for emotion classification.

- **Spatial adaptation:**

$$z_\ell^{(S)} = \text{Adapter}_S \left( \text{MSA} \left( \text{LN}(z_{\ell-1}) \right) \right) + z_{\ell-1} \tag{18}$$

- **Temporal adaptation:**

$$z_\ell^{(T)} = \text{Adapter}_T \left( \text{MSA} \left( \text{LN}(z_T) \right) \right) \tag{19}$$

- **Joint adaptation:**

$$z_\ell = \text{MLP} \left( \text{LN}(z_\ell^{(T)}) \right) + s \cdot \text{Adapter}_J \left( \text{LN}(z_\ell^{(T)}) \right) + z_\ell^{(T)} \tag{20}$$

Where $s$ is a scaling factor to control the weight of the output from Adapter.
The final video embedding $\mathbf{v} \in \mathbb{R}^D$ is computed by averaging the [CLS] tokens across frames:

$$\mathbf{v} = \frac{1}{T} \sum_{t=1}^{T} z_t^{[\text{CLS}]} \tag{21}$$

## H.2 TEXT ENCODER

We use the CLIP text encoder Radford et al. (2021) adapted with T2L Ahmad et al., which modifies each self-attention weight matrix by injecting trainable low-rank projections for efficient fine-tuning, as shown in the $l$-th transformer block in Fig. 12(b).

$$W_q \leftarrow W_q + A_q B_q, \quad A_q \in \mathbb{R}^{D \times r}, \quad B_q \in \mathbb{R}^{r \times D} \tag{22}$$

where $r \ll d$. Only $A_q$ and $B_q$ are trainable, enabling efficient adaptation. The final embedding is extracted from the [EOS] token:

$$\mathbf{t} = f_{\text{text}}(\mathcal{T}) \in \mathbb{R}^D \tag{23}$$

## H.3 AUDIO ENCODER VIA PRETRAINED HUBERT

Raw audio waveform $\mathbf{x} \in \mathbb{R}^S$, sampled at $16\,\text{kHz}$, is padded or truncated to a fixed duration of $S = 240{,}000$ samples (15 seconds). The waveform is then fed into a pretrained HuBERT-Base encoder Hsu et al. (2021) $\phi_{\text{HuBERT}}$ to extract frame-level speech representations:

$$\mathbf{H} = \phi_{\text{HuBERT}}(\mathbf{x}) \in \mathbb{R}^{T \times 768} \tag{24}$$

where $T$ is the number of time steps and 768 is the dimensionality of HuBERT-Base hidden representations. These frame-wise features are mean-pooled across the temporal dimension and passed through a lightweight projection head $f_{\text{proj}}$:

$$\mathbf{a} = f_{\text{proj}} \left( \frac{1}{T} \sum_{t=1}^{T} \mathbf{H}_t \right) \in \mathbb{R}^D \tag{25}$$

where $f_{\text{proj}}$ is a two-layer MLP mapping from 768 to $d = 512$ dimensions. The final embedding $\mathbf{a}$ is used as the audio representation.

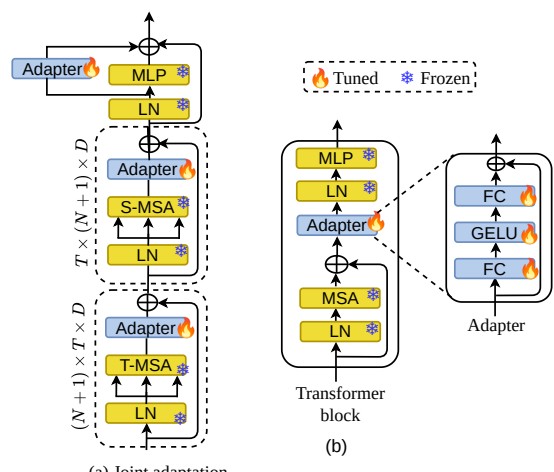

(a) Joint adaptation

Figure 12: Architecture of the video and text encoder modules used in SpEmoC. Module (a) depicts the $l^{\text{th}}$ block of the video encoder, which uses a transformer-based approach with temporal shift operations to capture spatio-temporal dependencies in frame embeddings, adapted from the AIM framework Yang et al. (2023). Module (b) shows the $l^{\text{th}}$ transformer block of the text encoder, adapted with low-rank projections (T2L) Ahmad et al. for efficient fine-tuning, where only the adapter parameters are updated during training, while other layers remain frozen.

# I TRAINING CONFIGURATION AND HYPERPARAMETERS

We utilize ViT-B/16-based CLIP as the visual encoder, extracting 8 sparsely sampled frames per video at 224×224 resolution, while audio features are derived using the HuBERT-Base model from `torchaudio` on raw 16 kHz audio with a maximum clip length of 15 seconds, where the HuBERT outputs (768-dim) are mean-pooled and projected to 512 dimensions via a two-layer MLP. For the training setup, we employ the AdamW optimizer with a learning rate of $5 \times 10^{-6}$, a weight decay of 0.2, a cosine decay scheduler with a 5-step warmup, and train for 50 epochs using a batch size of 20 on a single NVIDIA RTX A6000 GPU.

**Model Parameters:**

Table 25: Trainable parameters of each component in our model.

| Component | Trainable Parameters |
|---|---|
| CLIP Visual Encoder | 8.681M |
| Audio Projection MLP | 0.656M |